# Modular Micro Raman Reader Instrument for Fast SERS-Based Detection of Biomarkers

**DOI:** 10.3390/mi13101570

**Published:** 2022-09-22

**Authors:** Jamison Duckworth, Alexey V. Krasnoslobodtsev

**Affiliations:** 1Department of Physics, University of Nebraska Omaha, Omaha, NE 68182, USA; 2nDETKT, LLC, Omaha, NE 68104, USA

**Keywords:** Raman scattering, biomarker detection, surface-enhanced Raman scattering (SERS), automation, disease diagnostics, iSERS assay

## Abstract

Sensitive detection of biomarkers is very critical in the diagnosis, management, and monitoring of diseases. Recent efforts have suggested that bioassays using surface-enhanced Raman scattering as a signal read-out strategy possess certain unique beneficial features in terms of sensitivity and low limits of detection which set this method apart from its counterparts such as fluorescence, phosphorescence, and radiolabeling. Surface-enhanced Raman scattering (SERS) has also emerged as an ideal choice for the development of multiplexed bioassays. Such promising features have prompted the need for the development of SERS-based tools suitable for point-of-care applications. These tools must be easy to use, portable, and automated for the screening of many samples in clinical settings if diagnostic applications are considered. The availability of such tools will result in faster and more reliable detection of disease biomarkers, improving the accessibility of point-of-care diagnostics. In this paper, we describe a modular Raman reader instrument designed to create such a portable device suitable for screening a large number of samples with minimal operator assistance. The device’s hardware is mostly built with commercially available components using our unique design. Dedicated software was created to automatically run sample screening and analyze the data measured. The mRR is an imaging system specifically created to automate measurements, eliminating human bias while enhancing the rate of data collection and analysis ~2000 times. This paper presents both the design and capabilities of the custom-built modular Raman reader system (mRR) capable of automated and fast measurements of sandwich immunoassay samples on gold substrates using modified gold nanoparticles as Raman tags. The limit of detection (LOD) of the tested MUC4-specific iSERS assay was measured to be 0.41 µg/mL.

## 1. Introduction

Biomedical research and clinical diagnostics heavily rely on the availability of high-throughput diagnostic tools. In most cases, such tools are expensive and involve the use of research-grade devices requiring highly trained personnel for their operation. It is essential, therefore, to design and develop easy-to-operate portable technologies which will improve the accessibility of high-quality diagnostics to a broader population. Such technologies should provide a user-friendly and reliable usage of bioassays and biosensors at the point of care (POC).

The use of surface-enhanced Raman scattering (SERS) in sensing applications, such as bioassays and biosensors [1], has recently received considerable recognition as a highly sensitive platform for the detection of biomarkers in various kinds of biofluids including blood [2], urine [3], and saliva [4,5]. Particularly interesting are applications which have been reported in the fields of analytical studies, clinical diagnoses, environmental monitoring, and biomolecule detection [6]. The following are the beneficial features that attracted attention to using SERS-based bioassays: strong enhancements of Raman signals for molecules adsorbed on nanostructures made of noble metals such as silver and gold dramatically increase Raman scattering cross-section of molecular Raman tags [6,7,8]. The SERS effect overcomes the inherent weakness of Raman spectroscopy, which is the low intensity of Raman signals [9]. This effect combined with the characteristic molecular fingerprint of the Raman spectrum makes SERS a very promising candidate for various applications. Other beneficial features of the SERS readout methodology include narrow spectral bandwidth and the ability to perform multiplexed analysis of several markers using a single excitation wavelength [7,8,10]. 

Recent nanotechnological developments allow for more controllable manufacturing of nanostructures, significantly contributing to the popularity of SERS in various strategies aimed at the detection of molecular biomarkers. The rapid development of various SERS-based immunoassays has demanded the development of easy-to-use, portable, cheap, and automated tools for monitoring/reading a large number of samples simultaneously in research labs and clinics, especially if these assays are used for diagnostic purposes. 

Despite certain limitations, the SERS-based immunoassay platforms have superior capabilities of SERS readout strategy such as high sensitivity and simultaneous detection of a multitude of biomarkers. These advantageous capabilities can enable a direct, reliable, and affordable detection of low levels of biomarkers in various systems. Importantly, recently developed strategies improving the reliability and sensitivity of SERS-based immunoassays for low-level biomarker detection suggest that SERS can be effectively utilized for early diagnostics and disease monitoring. Early diagnosis and management of a disease pose the need for frequent and reliable testing. Frequent testing, thus, requires the availability of affordable and easy-to-use tools. Fast and easy-to-use diagnostic tools will improve the volume of point-of-care screening, substantially decreasing the economic burden. 

This paper describes the development of an accessible, user-friendly, and easy-to-use portable modular Raman reader (mRR) suitable for use at the point of care. It represents a unique approach to automated measurements of disease biomarkers using Raman spectroscopy. The mRR is a custom-designed imaging system specifically created to automate measurements, increase the rate of diagnostics, and decrease human bias. The modular design of the mRR allows for a great deal of flexibility to tailor capabilities to the specific needs of the user: speed, number of samples, and slides needed to be analyzed concurrently. The software is custom built in the LabVIEW environment (National Instruments, Austin, TX, USA), and the hardware components are mostly commercially available. The design and capabilities of the micro Raman reader are presented and described. The demand for portable and handheld Raman spectrometers has recently increased in a dramatic way due to the applicability of Raman spectroscopy in a variety of industries, including the agriculture [11,12,13], food [14,15,16], and pharmaceutical [17,18] industries. mRR design specifically focuses on the measurements for the SERS-based biomarker assays occupying a niche between handheld and research-grade instruments providing capabilities suitable for POC, such as automation of data collection and analysis.

The system is capable of performing automated and fast measurements of pre-prepared sandwich immunoassay samples utilizing gold microscope slides with a standard size of 75 by 25 mm. Automation of the sample movement and data analysis provides an advantage of hands-free operation. The automation of data acquisition and data analysis incorporated in the mRR increased the performance efficiency by a factor of ~2000, opening up the prospects to screen a large number of clinical samples. The methodology section covers the system’s hardware and software modules utilized in the design. We also describe the system’s setup and operation, data acquisition, and data analysis, as well as the efficiency, relevant statistics of collected Raman signals, and determination of the limit of detection on the example of MUC-4 cancer biomarker used in SERS-based sandwich immunoassay [10].

## 2. Materials and Methods

The following is the list of commercially available parts used in the design and assembly of the Raman reader (Figure 1 and Appendix A).

### 2.1. Hardware

Laser system: The laser operates at 647 nm wavelength and is from Innovative Photonic Solutions, IO647MM0150MF. Its full width at half maximum (FWHM) or the spectral width is <0.15. The maximum operational power of the laser is 150 mW. FC/PC for the output connection allows for connecting a variety of cables. 

Coupling optical fiber cable: The 647 nm excitation line is delivered using the fiber optics cable (Raman-Probe-647: from StellarNet, Inc., Tampa, FL, USA) which integrates both excitation and collection cables. The excitation fiber connects the laser head to the tip of the Raman probe, and the collection fiber connects the tip of the Raman probe with the spectrometer. The laser is connected via FC/APC connector, and the spectrometer is connected connect via SMA905. The core diameter of the collection fiber optics to the spectrometer is 600 μm, which is expected to provide 2 to 3 times the signal when compared to typical commercial probes. The Raman probe has integrated filters for the 647 nm laser line (with O.D. > 6) and a notch filter to remove quartz spectral contributions.

Microscope: The frame of the Olympus BX43 microscope is utilized (Olympus Corporation of Americas, Center Valley, PA, USA). The laser light is then passed through an Olympus PlanC N 40x objective (0.65 NA, FN 22, with working distance WD = 0.6 mm). An Olympus PlanC N 10x objective (0.25 NA, FN 22, with working distance WD = 10.6 mm) is used to visualize the position of the laser spot for the initial alignment of the system. Both objectives are mounted on a rotating turret–objective lens holder.

Translation stage and stage controller: The 2-dimensional stage model MLS203 (Thorlabs, Inc., Newton, NJ, USA) is the Fast XY Scanning Stage capable of translating up to 250 mm/s, which allows for movement in the x- and y-axes. The stage was equipped with the MLS203P10 multislide holder (Thorlabs, Inc., Newton, NJ, USA) allowing for simultaneous holding of four standard microscope slides (75 mm by 25 mm) for imaging. The translation range of the stage, 75 mm by 110 mm, conveniently allows for four slides to be measured in one session. Thorlabs’ BBD202 Controller (Thorlabs, Inc., Newton, NJ, USA) was used to allow for talking between the computer and MLS203 stage. The BBD202 is the two-channel controller that features Thorlabs’ standard Advanced Positioning Technology (APT) control and programming interface, enabling easy integration into automated microscopy applications. Additionally, for manual control of the XY stage, Thorlabs MJC001 Joystick was utilized. The stage is mounted on an Olympus BX43 microscope frame using a CSA1000-Fixed Arm Holder (Thorlabs, Inc., Newton, NJ, USA). An estimated cost of the system is under USD 15,000, which is more expensive than handheld Raman readers currently offered by several manufacturers, for example, Anton Paar (Graz, Austria), AvantesUSA (Lafayette, CO, USA), and even StellarNet Inc. (Tampa, FL, USA), whose spectrometer we used as the base for the mRR design. Most systems offer 532 nm, 785 nm, or 1064 nm excitation wavelength. mRR is custom-built for the excitation wavelength of 647 nm which is the best match for plasmon resonance of 60 nm gold nanoparticles (AuNP, NanoComposix, San Diego, CA, USA) utilized in the MUC4-iSERS assay [10]. In addition to portability, the mRR design also provides scanning capabilities, automation of data collection, and data analysis offering a smaller price tag than commercially available research-grade systems with scanning capabilities, such those from Horiba (Kyoto, Japan) or Renishaw (Wotton-under-Edge, UK). 

### 2.2. Making Addresses with a Polymer Stamp

In a typical preparation, the sandwich immunoassay modification procedure involves following several steps [10,19]: (1) stamping pattern of addresses with a stamp made out of polydimethylsiloxane (PDMS) (from Fisher Scientific Co LLC, Atlanta, GA, USA), (2) modification of gold substrate with a 3,3-dithio-bis-(succinimidyl)propionate (DSP) (Sigma Aldrich, Inc., St. Louis, MO) linker molecule, (3) binding of capture antibody (anti-MUC4 antibody 8G7 [20]) to DSP-modified surface overnight, (4) washing with phosphate-buffered saline (PBS) (Sigma Aldrich Inc., St. Louis, MO, USA) and blocking of the modified substrate with BSA for 4 h, (5) substrate binding of patient serum/standard samples overnight, (6) washing 4 times with PBS, (7) SERS label binding for 2–3 h, (8) washing 4 times with PBS, (9) (automated) data acquisition and (automated) data analysis. The process of the sample preparation has been previously detailed in [10,19]. Here we only focus on procedures critical for automation of assay analysis—Raman measurements and spectral analysis. 

The procedure starts with stamping of the gold-coated microscope slide (from Deposition Research Lab, Inc., St. Charles, MO, Ti/Au thickness ~40 nm/100 nm) using 1-octadecanethiol (ODT) (from Sigma Aldrich, Inc., St. Louis, MO) to create a pattern of addresses. A stamp is made using PDMS which is polymerized to make a thin flexible sheet—the thickness was chosen to be 5 mm to create both a durable and flexible stamp. Next, multiple holes of 3 mm in diameter are carefully punched in a 4 by 12 pattern keeping the same distance in the x-direction and the same distance in the y-direction between the addresses. The x- and y-directions do not necessarily have to be equal as this parameter is controlled by software. For the purpose of this study, the pattern was created to produce the same 5.5 mm separation distance between the addresses in both directions to minimize possible spillovers from one address to another during immunoassay preparation. The software is flexible and is able to handle any rectangular pattern of addresses for the user’s convenience. The program allows a user to enter the following input parameters to define a scanning pattern: (1) the number of addresses in the pattern, (2) offset distance defining the position of the first address relative to the edge in both x- and y-directions, (3) distance between addresses in the x-direction, and (4) distance between addresses in the y-direction. 

### 2.3. Software and Spectral Imaging

LabVIEW (National Instruments, Austin, TX, USA) visual programming language was used to create a home-built user-friendly interface for both controlling the 2-dimensional stage (MLS203 from Thorlabs, Newton, NJ, USA) and controlling data collection via Raman-HR-TEC (StellarNet, Inc., Tampa, FL, USA). The choice of this particular software was justified by its ability to handle flow control easily and quickly. The imaging is performed by applying a combination of a raster scan (address to address), as shown in Appendix A, and a vector scan (within the address), as shown in Appendix A. The raster scan starts at the bottom left of the slide and transitions right for a preset distance, up another preset distance, left by the first preset distance, and up again by the second preset distance (Appendix A). All preset distances can be selected by a user before the start of the scan depending on the pattern of the stamp. Both preset distances d1 and d2 were equal to 5.5 mm for our rectangular pattern. The vector scan, scan within the address, and images in concentric circles are shown in Appendix A. The minimum radius for the circle is 0.15 mm, and thus the distance between concentric rings is 0.3 mm, while the maximum radius is that of the sample. Step size for each ring is calculated as the diameter as well as the circumference. This ensures no overlap of inner-address scanning. Raman spectrum is acquired after the stage is moved and parked in the next spot. The process continues until the preset number of scans (six in this study) is measured within an address, after which the stage is moved to the next address, returning to the raster scan (Appendix A). The process is repeated until all of the addresses have been probed. After completing the run that sweeps out the area of the slide, the program returns the 2-dimensional scanning stage back to the start and will proceed to the next slide if the option of multislide scanning was chosen. If the multislide option was not selected, the program will stop the scanning run.

## 3. Results

This section describes the setup of the modular micro Raman reader (mRR) assembly and its performance. The initial motivation for constructing the Raman reader was to build an automated system on a budget for samples with Raman readout and suitable for SERS-based detection platforms. The system was designed to have capabilities and advanced automation features where the screening of a large number of samples is of critical importance. The built-in advanced features provide portability which is easy to use. The design also allows for a modular setup customizable for various sample preparation procedures and a budget price. The fully assembled Raman reader is suitable for automated screening of samples prepared using a sandwiched immunoassay. The details of sample preparation are described elsewhere [10]. However, it is important to note that the samples are prepared on a gold-coated microscope slide with a typical size of 25 mm by 75 mm, thus fitting onto the MLS203P10 multislide holder. In addition, each sample is a small round spot—address—of about 3 mm in diameter. More importantly, the addresses are prepared in a certain pattern that could be programmed in the software to be scanned in a consecutive manner (see section “Stamp and Addresses” for details). 

### 3.1. Raman Reader Imaging Setup

The imaging setup is schematically shown in Figure 1. The system is configured around an Olympus BX43 microscope (Figure 1, I). The laser operating at 647 nm (Figure 1, II) is connected via optical fiber (Figure 1, III) into the Raman probe (Figure 1, IV) to the top of the microscope for Raman imaging. The top of the microscope is fitted with a Raman probe holder (Figure 1, V) that features a multiscrew system for holding the tip of the Raman probe in place and is also suitable for adjusting the optimal vertical alignment of the laser beam with the microscope objective. The laser light is passed through an Olympus Plan C N 40x objective (Figure 1, VI) and focused onto the sample (Figure 1, VII). Samples are patterned with the PDMS stamp (see section “Sample Pattern”) onto a gold-coated microscope slide [10], which is held by the MLS203P10 multislide holder. The holder is fitted into the two-dimensional fast XY scanning motorized stage (Figure 1, VIII) capable of translating up to 250 mm/s allowing for movement in the *x*- and *y*-axes with translation ranges of 110 mm by 75 mm, respectively. The stage is mounted on the Olympus BX43 frame using the fixed arm holder that then allows using the microscope’s focusing wheel to adjust the z-axis position of the samples for the optimal read-out intensity matching the working distance of the objective lens. Although it is possible to assemble the Raman reader using a three-axis motorized stage, thus allowing for automated z-adjustment, we have resorted to manual adjustment of z-position to lower the cost of the system. Additionally, the z-position only needs to be adjusted once for multiple measurements with the proper horizontal alignment of the stage. The stage is controlled with Thorlabs’ BBD202 Controller (Figure 1, IX) making use of two available channels to control x- and y-directions (the third channel could be used for z-axis adjustment if necessary). We also equipped our system with Thorlabs’ MJC001 Joystick (Figure 1, X) for manual control of the XY stage position, useful for the initial optimization of the system’s parameters. The operation of the Raman reader system is performed from a PC (Figure 1, XI). Two critical units, namely the (1) Raman spectrometer (data collection unit (Figure 1, XII)) and (2) BBD202 Controller (positioning unit (Figure 1, IX)), are both connected to the PC using the USB interface and controlled with a dedicated software package custom-built using the LabVIEW environment.

### 3.2. Software

For controlling data acquisition and positioning of the two-dimensional stage via computer, the LabVIEW environment was used. LabVIEW (National Instruments, Austin, TX, USA) was chosen for its ability to handle flow control easily and quickly, capable of integrating (1) the SpectraWiz to control Raman spectrometer as a subVI (from StellarNet, Inc., Tampa, FL, USA) and (2) Advanced Positioning Technology (APT) control software suite to control the BBD202 unit (from Thorlabs, Newton, NJ, USA). The complete logic flow as well as essential parts of the code are provided in the Appendix A. The following are features built into the software to provide extra flexibility and additional imaging options: (1) automated imaging of the predefined pattern of stamped addresses; (2) imaging across a single address; and (3) single spot time series. The specialized mRR software controls both the movement of the motorized stage and Raman spectra acquisition. Thus, automated measurements are possible with minimal engagement of the instrument’s operator which is only required to set up initial parameters for data acquisition. Automated data acquisition contributes to the overall goal of making rapid high-throughput measurements.

### 3.3. Raman Reader Operation

#### 3.3.1. Data Acquisition

The custom-built Raman reader incorporated into an Olympus BX-43 microscope frame also accommodates a PC-controlled motorized XY stage to help perform automated measurements for a large number of samples in a high-throughput assay. Attached to the center of the stage is MLS203P10 multislide holder, allowing for four slides (25 mm by 75 mm) to be held in place for imaging. This set of hardware makes automated measurements possible for four slides concurrently in one run. To ensure that the programmed stage will hit the modified spots on a slide, a PDMS stamp was made for address deposition (see “Address Pattern and PDMS stamp” section). A flexible option was built into the software program to accommodate any pattern of the stamp design, thus adding flexibility to a measurement procedure for various immunoassay modifications. For the purpose of this study, the stamp had address holes placed in a 4 by 12 pattern that were equidistantly spaced across the 25 by 75 mm slide, Appendix A, making a total of 48 addresses. The distance has been entered into the program to unambiguously identify the locations of the modified addresses. Since the motorized stage can measure four slides concurrently, it increases the capacity to 192 samples per reading. Miniaturization of the addresses is another option to accommodate an even larger number of addresses on a single slide. Such miniaturization is possible with robot-assisted sample deposition and the stage’s superb performance and characteristic high lateral resolution (3 µm accuracy and 0.25 µm repeatability) as long as a pattern is stamped reproducibly.

Automation of the measurements contributes to the overall streamlining of data acquisition. A rough estimate suggests that an operator would average about three scans every 10 min without automation, and manual positioning also would contribute to inaccuracy. The automation process increases accuracy and the output to 12 scans per minute (for 5 s acquisition time per spectrum). A simple estimate suggests that in one hour, the efficiency increases from around 18 scans to 720 scans, which is ~40 times more efficient.

The immunoassay procedure begins with stamping of the gold slide with ODT within the pattern defined by the PDMS stamp followed by the immunoassay preparation (Figure 2A) [10]. The pattern of the stamp defines the locations of the addresses where the localized sandwich immunoassay provided binding of antigen and Raman tags (Appendix A) [7,10]. The scanning then follows the pattern of the stamped addresses, only measuring the distribution of Raman tags within the modified immunoassay addresses. Immunoassay assembly is shown in Figure 2A with several critical components: (1) capture substrate (circled in blue), (2) AuNP Raman tags (circled in red), (3) hardware and software (circled in green). This paper describes the operation of the Raman reader which provides hardware and software for controllable excitation and automated data acquisition and analysis.

Spectral data are typically acquired as Raman intensity versus wavenumber (Figure 2C). We have also built the automated data analysis of the saved Raman spectra to compute intensities of the Raman signals of interest for well-separated Raman peaks. The test of the instrument performance provided in this study involved the use of 4-nitrobenzene thiol as a Raman tag molecule, Figure 2B, and hence the 1336 cm-1 peak corresponding to the symmetric stretch of the nitro group (-NO2) was utilized for testing the performance of the Raman reader. The end result of data analysis is the background-corrected intensity (Appendix A) reporting the local concentration of the biomarker (antigen) captured by the surface-immobilized antibodies in the sandwich immunoassay [7,10,19]. The spectra are smoothed using a five-point quadratic Savitzky–Golay filter, which conserves the trend while increasing the signal-to-noise ratio and is then passed through a three-point triangle smoothing function. Next, the program selects the main peak based on the mean value of the intensity. The program returns the intensity found by subtracting the baseline from the value of the main peak’s intensity, shown in Figure 2. Since several spectra are recorded for each address for statistical purposes, intensities are calculated for each spectrum measured within an individual address and averaged. This analysis is also automated using the LabVIEW platform, allowing data acquisition to be streamlined with data analysis. Automation of data analysis increases efficiency in data acquisition by an estimated factor of ~500. Such an algorithm can be utilized to compute intensities for other Raman peaks in, for example, multiplexed assays.

#### 3.3.2. Evaluation of Sample Heterogeneity within a Modified Address

With complete automation of the Raman measurements, it became possible to establish homogeneity of immunoassay modification within a single address. The Raman intensity of the 1336 cm^−1^ band which indicates efficient binding of the Raman tags to the surface was measured across a round address along its diameter with the step size of 5 μm. Figure 3 shows representative profiles of the Raman intensity variation when an address is scanned for a typical round address used in a sandwich immunoassay. The profile indicates that some areas exhibit larger binding efficiency of the antigen to the capture surface. The relative homogeneity of the sample coverage as indicated in Figure 3 justifies our method of vector scanning within the address where only several spots are measured and averaged to provide a single value of Raman intensity for each address. This method of averaging random spots only also contributes to a faster acquisition process due to the smaller amount of spectral information required.

#### 3.3.3. Evaluation of Statistical Significance in Automated Data Acquisition

We have also analyzed the statistical significance of SERS measurements within a spot. The heterogeneous nature of sample coverage requires obtaining an average measured value and providing standard deviation (or standard error of the mean). Figure 4A shows a plot that represents how the value of averaged Raman intensity changes upon the addition of extra measurements from minimum 3 to maximum 60. While a very small number of measurements is not enough for reporting averaged Raman intensity reproducibly as one can see from the graph, we start oversampling at a sufficiently large number of measurements. The value of averaged Raman intensity does not change, and the standard deviation remains the same for the averaged total measurements above ~16. An additional random test of single point removal from statistical analysis supports our conclusion that 16 measurements are the required minimum to build a statistically significant population with minimal deviation of the mean and error. Figure 4B shows three series of random point removal, and all three show that the averaged values start diverging significantly at below ~16 averaged measurements. Therefore, a minimum of 16 measurements works well for the analysis of samples using the automated data acquisition mode of the micro Raman reader.

#### 3.3.4. Evaluation of SERS Signal Stability

The typical SERS-based immunoassay involves the use of SERS tags constructed from gold nanoparticles (AuNPs) and Raman active molecules to produce a read-out signal whose intensity reports the amount of analyte in a sample [7,10]. Many studies have indicated that a careful choice of the acquisition parameters is required to reproducibly measure the SERS signal, which is very important in utilizing such SERS-based immunoassays for analytical purposes [7,10]. The intensity of the Raman signal may fluctuate and deteriorate, making it difficult to evaluate concentrations of a disease biomarker. An example of such deterioration is shown in Figure 5A for 1336 cm^−1^ of the symmetric nitro stretch in the NBT molecule. There are two major contributors to the SERS intensity changes: (1) physical, i.e., photothermal deactivation of Raman active molecules and their desorption from the surface, and (2) chemical, due to photo-driven catalytic conversion of molecules to other species. The first effect is manifested by a gradual decrease in Raman intensity, and the second effect results in the appearance of new Raman peaks, a decline in intensity, and a shift in the position of the original peaks. Several protection strategies have been developed to reduce the effects associated with the signal decline. The strategies include inert layers of inorganic material (silica), transparent polymer films, and crowns of polyethylene glycol molecules around gold Raman tags [7,10]. Careful choice of parameters such as laser power and the length of exposure may eliminate the need for the labor-intensive preparation of samples with any kind of protection. The necessity for an initial evaluation of the parameters and their optimization prompted us to introduce an extra feature into the Raman reader—the evaluation of signal deterioration rate. A quick analysis of a single spot on an address reports back on whether the set of initial parameters is optimal for the measurements. Figure 5B shows three curves recorded on the sample at three different laser intensities, (I) 100 mW, (II) 125 mW, and (III) 150 mW of the output power. The larger output power results in a faster signal decline, while lower power and in turn smaller LSPR effect shows a more gradual decline. These results indicate that 100 mW will be a better choice of the output laser power for recording the spectra with longer acquisition times. Therefore, using the automatic feature incorporated into mRR allows for quick evaluation of signal stability and consequently adjustment of the parameters such as acquisition time and laser power to obtain more reliable and reproducible readings of the evaluated assay.

#### 3.3.5. Assay Calibration and Assessment of Limit of Detection

The multiple steps of the sandwich SERS-based immunoassay are [10]: (1) creating a pattern of addresses with PDMS stamp and ODT, (2) modification of gold surface within the addresses using DSP, (3) capture antibody binding to DSP-modified surface followed by multiple washes with PBS and blocking of the substrate with BSA, (4) substrate binding of patient biomarker from the samples followed by multiple washes with PBS, (5) SERS tag binding and multiple washes, (6) (automated) data acquisition and (automated) data analysis.

Multiple steps of the procedure introduce multiple points of potential error requiring calibration of the assay’s binding capability and determining the limit of detection (LOD). For these purposes, protein lysate from pancreatic cancer CD18 cells was used as an internal control in a dilution series. A calibration curve (CD18 protein lysate) is typically constructed using a series of dilutions with the following concentration range: 100, 50, 18.75, 6.125, 3.13, 1.56, and 0 µg/mL. Figure 6 presents one such typical calibration curve showing a gradual increase in SERS intensity measured on the prepared addresses of the immunoassay using the mRR instrument. The calibration curve is the plot of Raman intensity signals versus lysate concentration. The dependence is then typically fitted with the following equation:I(C_X_) = (B_MAX_ C_X_)/(K_D_ + C_X_) + N_0_,(1)

N_0_ is the background signal that coincides with the blank sample in the measurements. I(Cx) is the intensity of the signal at antigen concentration Cx (lysate concentration), B_MAX_ is the maximum binding capacity, and K_D_ is the apparent dissociation constant. Typically, the curve has a sigmoidal shape with an apparent maximum corresponding to the assay’s capture saturation (Figure 6). The limit of detection (LOD) for the assay is defined as the background signal plus three standard deviations of the intensity of the blank measurement.
LOD = N_0_ + 3σ,(2)

Table 1 shows parameters obtained from the fit using Equation (1). Based on the fit of the calibration points, the limit of detection was determined to be LOD = 1800 counts of spectral intensity translating into 0.41 µg/mL of MUC4 concentration.

## 4. Conclusions and Outlook

The development of portable SERS-based readers may play a significant role in the application of SERS-based immunoassays for early diagnostics, clinical translation, and point-of-care improvement. The development of such a system requires a thorough design to simplify data acquisition and data analysis and be user-friendly to reduce operator efforts. We have designed, assembled, and tested a portable Raman reader device capable of automated analysis of microscope slides for the SERS-based immunoassays (iSERS-mRR). The design of the portable Raman reader (mRR) is modular and can be separated into five different parts for portability. Modular design allows the user to choose an appropriate assembly for a specific need of the user. Automation, data acquisition, and data analysis are all included through LabVIEW and coupling to pre-existing code. A more efficient algorithm was achieved by allowing a continuous scanning method. The scanning method introduces concentric circle scanning and gives a reasonable homogeneous scanning pattern for data around a central point. The minimization of sample addresses should further improve streamlining and contribute to high-throughput screening of clinical samples from patients. We believe that overcoming the complexity of the Raman signal recording devices and further improvements in SERS-based immunoassay preparation procedures coupled with the Raman reader presented here shall result in simple, rapid, and inexpensive diagnostic capabilities of great importance for low-resource point-of-care settings.

## Figures and Tables

**Figure 1 micromachines-13-01570-f001:**
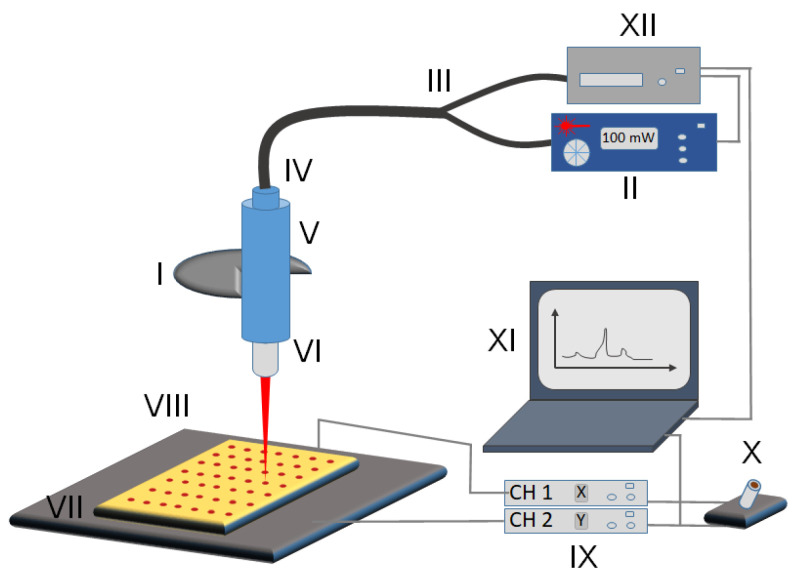
Schematic of the micro Raman reader imaging setup. (I) Objective holder of the microscope frame, (II) laser controller, (III) optical fiber, (IV) Raman probe, (V) Raman probe holder, (VI) laser objective, (VII) sample slide, (VIII) 2-dimensional translatable stage, (IX) BBD202 stage controller with x and y channels separately, (X) joystick for manual control of the stage, (XI) computer controlling both spectrometer and stage, (XII) Raman-HR-TEC spectrometer.

**Figure 2 micromachines-13-01570-f002:**
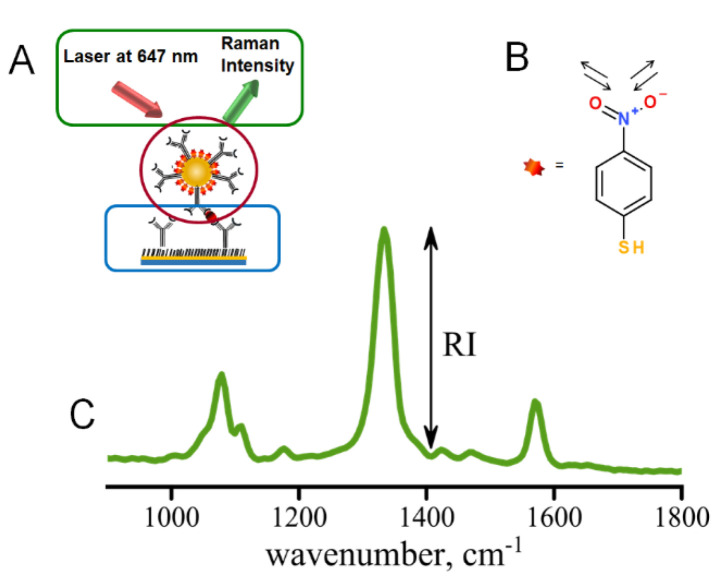
(**A**) Sandwich immunoassay involving capture substrate modified with specific antibodies (blue), Raman tags (modified AuNPs) to carry secondary antibodies (red), excitation and data collection provided by the mRR (green); (**B**) structure of the NBT reporter molecule with intense Raman band characteristic of nitro stretch—at 1336 cm^−1^; (**C**) typical Raman spectrum observed for immunoassay using NBT-modified AuNPs as Raman tags.

**Figure 3 micromachines-13-01570-f003:**
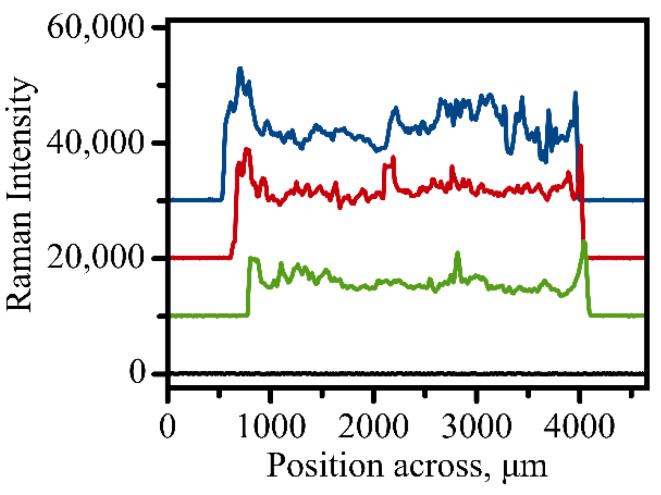
Three representative scans of Raman signal variation when a 3 mm address is scanned across utilizing automated measurements. Intensity was evaluated for the 1336 cm^−1^ peak of the NBT molecule. Black line shows background signal variation measured on an empty unmodified spot on the golden slide.

**Figure 4 micromachines-13-01570-f004:**
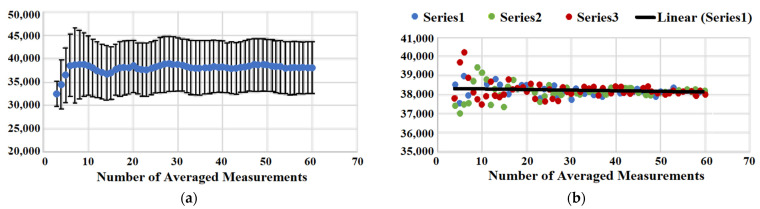
(**a**) Averaged signal intensity (blue points) as a function of averaging the number of measurements and corresponding standard deviation (error bars). (**b**) Three series of random single point removal. Averaged SERS intensity is plotted vs. number of averaged measurements.

**Figure 5 micromachines-13-01570-f005:**
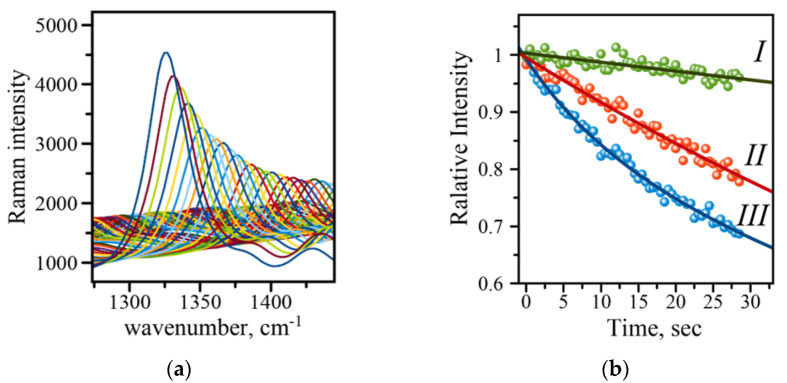
(**a**) Raman intensity decline with time demonstrated on an example of the 1336 cm^−1^ band of the symmetric stretch of the amino group in the NBT molecule (measured at 150 mW laser power; (**b**) comparison of intensity decline at three different output powers: (I) 100 mW (green), (II) 125 mW (red), (III) 150 mW (blue). Solid lines are exponential function fits to the data points recorded with 0.5 s acquisition time.

**Figure 6 micromachines-13-01570-f006:**
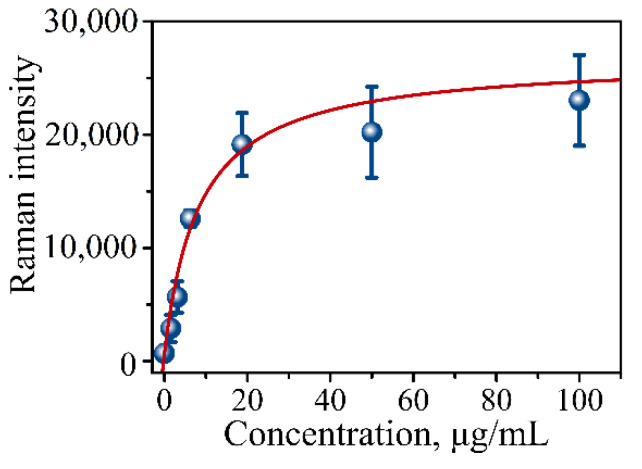
Calibration curve of CD18 protein lysate in serial dilutions measured using our established procedure for the immunoassay. Raman signal intensity was plotted against total μg/mL of CD18 lysate representing concentration of MUC4 at 0, 1.56, 3.13, 6.125, 18.75, 50, and 100 µg/mL concentrations.

**Table 1 micromachines-13-01570-t001:** Fitting parameters of the calibration curve in Figure 6 using Equation (1).

Parameter	Value
B_MAX_, counts	26,076
K_D_, µg/mL	8.4
N_0_, counts	612

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
