# Peer review of "Modular Micro Raman Reader Instrument for Fast SERS-Based Detection of Biomarkers"

_micromachines, 2022, doi:10.3390/mi13101570_

Round 1

Reviewer 1 Report

Duckworth and Krasnoslobodtsev presented "Modular Micro Raman Reader instrument for fast SERS-based detection of biomarkers." This study presents the creation of a simple and user-friendly portable modular Raman Reader (mRR) for use at the Point-Of-Care. It provides a novel way to measuring illness biomarkers using Raman Spectroscopy. The mRR is a custom-designed imaging system that was built to automate measurements, boost diagnosis rate, and reduce human bias. The mRR's modular architecture provides for a significant degree of flexibility in tailoring capabilities to individual user demands, such as speed, quantity of samples, and slides to be processed concurrently. The device is demonstrated to be capable of performing automated and quick measurements on pre-prepared sandwich immunoassay samples using gold microscope slides with standard dimensions of 75 by 25 mm. The manuscript is well-written and the results supports the claims of the authors. I have the following minor comments. 

*Authros should include the major findings in the abstract (figures of merits, detection limit, etc.)

*Estimated cost of the system could be useful for the readers.

*A comparision to commercial Raman systems interms of cost and performance could be useful. 

Author Response

We thank the reviewer for these suggestions. We have now included the limit of detection and the advantages of mRR including the increased speed of data collection in the abstract. We also included an approximate estimate of the cost for the system at the very end of section 2.1. A comparative example of commercially available scanning Raman systems is also added to section 2.1.

Reviewer 2 Report

The paper "Modular Micro Raman Reader instrument for fast SERS-based detection of biomarkers" show some interesting progress on the development of miniaturized Raman sensors. I would recommend the publication in "Micromachines" after the following corrections:

- There are a lot of statements on the introduction part which are not supported by appropriated references. Please add the references...

- It would be good to have a paragraph in the introduction explaining what is the state-of-the-art for portable Raman readers, and how the presented research goes beyond the state-of-the-art.

- In the methods section, a picture of the setup will allow the reader to understand the portability of the system.

Author Response

We would like to thank the reviewer for these valuable comments which we now addressed in the manuscript as detailed below:

1) We added appropriate references in the introduction part.

2) A brief description of portable and handheld Raman readers, what differentiates mRR from available Raman instruments, and a comparative example of cost with commercial scanning Raman instruments are now added at the end of section 2.1.

3) We do refer the reader to Figures 1 and S1 (supplement) showing the setup of the instrument. We shall leave it up to the editorial office to make appropriate arrangements for the figures in the final version of the paper.